# Novel Cyclic Peptides from Lethal *Amanita* Mushrooms through a Genome-Guided Approach

**DOI:** 10.3390/jof7030204

**Published:** 2021-03-11

**Authors:** Shengwen Zhou, Xincan Li, Yunjiao Lüli, Xuan Li, Zuo H. Chen, Pengcheng Yuan, Zhu L. Yang, Guohong Li, Hong Luo

**Affiliations:** 1Yunnan Key Laboratory for Fungal Diversity and Green Development, Kunming Institute of Botany, Chinese Academy of Sciences, Kunming 650201, Yunnan, China; zhoushengwen@mail.kib.ac.cn (S.Z.); lixincan@mail.kib.ac.cn (X.L.); lvliyunjiao@mail.kib.ac.cn (Y.L.); yuanpengcheng@mail.kib.ac.cn (P.Y.); fungi@mail.kib.ac.cn (Z.L.Y.); 2CAS Key Laboratory for Plant Diversity and Biogeography of East Asia, Kunming Institute of Botany, Chinese Academy of Sciences, Kunming 650201, Yunnan, China; 3School of Life Sciences, Yunnan University, Kunming 650091, Yunnan, China; ligh@ynu.edu.cn; 4University of Chinese Academy of Sciences, Beijing 100049, China; 5Department of Environmental Science and Engineering, Kunming University of Science and Technology, Kunming 650091, Yunnan, China; x.li@hotmail.com; 6College of Life Science, Hunan Normal University, Changsha 410081, Hunan, China; chenzuohong@263.net

**Keywords:** poisonous mushroom, genome, MSDIN family, Amanitin, LC–HRMS

## Abstract

Most species in the genus *Amanita* are ectomycorrhizal fungi comprising both edible and poisonous mushrooms. Some species produce potent cyclic peptide toxins, such as α-amanitin, which places them among the deadliest organisms known to mankind. These toxins and related cyclic peptides are encoded by genes of the “MSDIN” family (named after the first five amino acid residues of the precursor peptides), and it is largely unknown to what extent these genes are expressed in the basidiocarps. In the present study, *Amanita rimosa* and *Amanita exitialis* were sequenced through the PacBio and Illumina techniques. Together with our two previously sequenced genomes, *Amanita subjunquillea* and *Amanita pallidorosea*, in total, 46 previously unknown *MSDIN* genes were discovered. The expression of over 80% of the *MSDIN* genes was demonstrated in *A. subjunquillea*. Through a combination of genomics and mass spectrometry, 12 *MSDIN* genes were shown to produce novel cyclic peptides. To further confirm the results, three of the cyclic peptides were chemically synthesized. The tandem mass spectrometry (MS/MS) spectra of the natural and the synthetic peptides shared a majority of the fragment ions, demonstrating an identical structure between each peptide pair. Collectively, the results suggested that the genome-guided approach is reliable for identifying novel cyclic peptides in *Amanita* species and that there is a large peptide reservoir in these mushrooms.

## 1. Introduction

The genus *Amanita* (Persoon 1797) is double-faced: on one hand, it harbors some of the best-known gourmet mushrooms such as *Amanita caesarea*, once favored by emperors of Rome [1]; on the other hand, it causes over 90% of deadly mushroom poisonings worldwide because most have little knowledge to distinguish the edible from the deadly [2,3,4]. The majority of the species within this genus are important ectomycorrhizal fungi that have associations with more than ten families of trees, and they play important roles in the ecosystem health of forests [5].

Research on toxins in amanitin-producing (referred as deadly or lethal in this report) *Amanita* mushrooms dates back to the 19th century. Due to technical difficulties in extracting, purifying, and analyzing the chemical substances, significant advances were not made until the 20th century [6]. Lynen and Wieland purified one of the major toxins, phalloidin, in 1938 [7]. Then, α-amanitin was purified and crystalized three years later [8]. The death cap, *Amanita phalloides*, became well-known for its deadly poison in 1951 from death reports [9]. As research has progressed, two major types of toxins, i.e., amatoxins and phallotoxins, have been found in dozens of lethal *Amanita* species. The amatoxins function as a highly specific, efficient inhibitors of eukaryotic RNA polymerase II [10,11], and the phallotoxins effectively prevent the depolymerization of actin fibers [12,13,14]. It has been shown that amatoxins and phallotoxins in *Amanita* are synthesized on ribosomes, which represents the first ribosomal cyclic peptide discovered in the Fungi kingdom [15]. They are encoded by the “MSDIN” gene family as precursor peptides of 34–37 amino acids. The precursors are then cleaved and macrocyclized into 7–10 amino acid cyclic peptides by a specialized prolyl oligopeptidase B (POPB) [16,17]. It is a bifunctional enzyme catalyzing both the hydrolysis of peptide bonds and transpeptidation. To achieve mature toxins, most of these resultant cyclic peptides undergo further posttranslational modifications, mainly including multiple hydroxylations, sulfoxidation, epimerization, and the formation of a cross-bridge between Trp and Cys; the order in which these reactions occur is unknown. The *MSDIN* genes encoding amatoxins, such as α- and β-amanitin, and phallotoxins, such as phallacidin and phalloidin, are readily found in the sequenced genomes of lethal *Amanita* species [15,18]. In addition, roughly 20–30 uncharacterized *MSDIN* genes have been identified in each of these genomes [18,19]. In the cases of *Amanita bisporigera* and *A*. *phalloides*, about half of the unknown *MSDIN* genes have been found to be expressed at the transcription level, and the results of mass spectrometry have indicated that two of the genes produce corresponding cyclic peptides [18,19]. In the past 10 years, only a few novel cyclic peptides have been discovered, including amanexitide in *Amanita exitialis* [20], and cycloamanides F and E (CylF and CylE) in *A. phalloides* [19]. In total, roughly 25 cyclic peptides have been described [1,21]. In contrast, newly sequenced genomes of lethal *Amanita* species show that the *MSDIN* genes by far outnumber the count of known cyclic peptides, indicating there could be a much larger potential for novel cyclic peptides in these mushrooms. However, it is mostly unknown whether these *MSDIN*s are actually translated into cyclic peptides.

Conventional methods for cyclic peptide characterization in *Amanita* mainly use thin-layer chromatography (TLC), HPLC, hydrolysis, and NMR techniques, which generally need a significant amount of examined material [1,6]. Though somewhat biased due to monitoring wavelengths, HPLC analyses in many laboratories have suggested the most significant peptide production in *Amanita* species is attributed to the major toxins, i.e., α-amanitin, β-amanitin, phallacidin, and phalloidin. Noticeably, a number of less significant peaks are present in the vicinity of these known peptides, many of which are of insufficient amount for further analyses via conventional methods. Given the situation, a more sensitive method would be very appreciated.

Functionally speaking, cyclic peptides produced by *Amanita* species are more diverse than those commonly known from just the toxins. Antamanide, a cyclic decapeptide, protects mice from phalloidin toxicity. Through competition, the molecule effectively inhibits the uptake of phalloidin by the OATP1B1 (organic anion–transporting polypeptide family) transporter of the liver cells [22]; the same transporter is used by phallotoxins [23]. Newer findings suggest that the peptide is a novel inhibitor of the mitochondrial permeability transition pore [24]. Cycloamanides (CylA, CylB, CylC, and CylD), antamanide, and their synthetic derivatives display immunosuppressive activities in both in vivo and in vitro assays [25,26,27]. In *A. phalloides*, researchers suspect that CylE and CylF are related to cycloamanides due to sequence similarities [6]. These examples have placed many of the cyclic peptides outside the toxin category, indicating that they may have diverse functions in *Amanita* mushrooms.

In the present study, two lethal *Amanita* species, *Amanita rimosa* and *A. exitialis*, were sequenced, and their *MSDIN* genes mined. Together with two previous sequenced species, *Amanita pallidorosea* and *Amanita subjunquillea*, four sets of *MSDIN* genes were analyzed using a combination of methods. Linear and cyclic peptides were predicted by directly using amino acid sequences in the core regions of the *MSDIN* genes. Modifications after cyclization, such as hydroxylation, sulfoxidation, and cross-bridge formation, were taken into consideration. Possible molecular formulas with and without further modifications were generated for all candidate peptides. Through high resolution MS (HRMS), masses matching putative cyclic peptides were mined. Further amino acid composition was elucidated based on MS/MS results, which were manually analyzed via multiple peptide analyzing platforms. In addition, three synthetic peptides were obtained and included in this study for further structure confirmation.

## 2. Materials and Methods

### 2.1. Collection and Preservation of Samples

Fresh fruiting bodies of *A. subjunquillea* (*As*), *A. exitialis* (*Ae*), *A. rimosa* (*Ar*), and *A. pallidorosea* (*Apa*) were collected in the Yunnan, Guangdong, Hunan, and Shandong provinces of China, respectively. All the above samples were immediately submerged in dry ice on site, transferred to liquid nitrogen tanks the same day, freeze-dried, and then stored at −80 °C. All the mushrooms were taxonomically identified according to previous studies [3,5].

### 2.2. Genome and Transcriptome Sequencing

The sequencing platform for the genomes of *Ae* and *Ar* was PacBio Sequel at NextOmics Biosciences, Wuhan, China. Sequencing and assembly were carried out using the company’s standardized pipeline. High quality DNA was extracted as described before [18], and a 20-kbp library was constructed using a PacBio template prep kit and then analyzed by Agilent 2100 Bioanalyzer for quality control. After the completion of the library, the DNA template and enzyme complex were transferred to the Sequel sequencer for real-time single molecule sequencing. An Illumina HiSeq X10 platform was used for nucleotide level correction based on a 350-bp library and the company’s standard method. For the following analyses, two additional genomes of *As* and *Apa* were included from our previous studies [18].

One young fruiting body of *As* was chosen for transcriptome sequencing. The process was carried out via the standardized pipeline at NextOmics Biosciences, Wuhan, China. The library was constructed after the samples passed quality inspection. The complementary DNA (cDNA) library was constructed by magnetic bead enrichment. Clean reads were obtained from the Illumina RNA-Seq platform, these were then aligned to the *As* genome using the HISAT2 software [28]. Transcriptome assembly was performed using StringTie under default parameters [29].

### 2.3. Extraction of Cyclic Peptides

Freeze-dried material (0.05 g) for all samples was ground to fine powder with liquid nitrogen in a mortar and pestle. Then, a 1 mL extraction buffer containing absolute methanol:water:0.01 M hydrochloric acid (5:4:1) was added as described previously [30,31]. The mixture was heated in a water bath at 75 °C for 15 min before centrifugation at 12,000 rpm for 10 min. The supernatant was transferred to a new 1.5 mL centrifuge tube, filtered with a 0.22 μm polyethersulfone syringe filter (BS-PES-22, Biosharp), and then stored at −80 °C before use.

### 2.4. HPLC Analysis

An Agilent 1260 Infinity II HPLC system equipped with a UV detector (Agilent Technologies, Santa Clara, CA, USA) was employed for the analysis. Separations were carried out on an InfinityLab Poroshell 120 EC-C18 reverse phase HPLC column (4.6 × 100 mm I.D., particle size 2.7 μm, Agilent Technologies) at 28 °C. The mobile phases were (A) 0.02 M aqueous ammonium acetate–acetonitrile (90:10, *v*/*v*) and (B) 100% acetonitrile. The pH of solvent A was adjusted to 5.0 with glacial acetic acid. All solutions were prepared with HPLC-grade reagents and were degassed before use. For HPLC separation, the flow rate was 0.5 mL/min with constant 100% solution A for 2 min, followed by a gradient from 100% solution A to 100% solution B over an 8 min time course. The absorbance of the elution was monitored at 280 and 295 nm.

### 2.5. Mining MSDIN Genes

All nucleotide sequences of *MSDIN*s from their genomes and the *As* transcriptome were obtained through standalone Basic Local Alignment Search Tool (BLAST) searches at National Center for Biotechnology Information (NCBI BLAST + 2.6.0) with query *MSDIN* sequences from *Ab* (EU196139–EU196158) and *Galerina marginata* [32], which are well-characterized by our molecular and biochemical approaches [15,32]. In addition, *MSDIN*s with a high divergence at the core region from our previous reports were added as queries for a more thorough excavation of potential *MSDIN* genes [18].

### 2.6. Predicted Molecular Masses of Candidate Peptide Sequences

To obtain molecular masses of predicted peptides based on *MSDIN* sequences, the Molecular Isotopic Distribution Analysis (MIDAs) website (https://www.ncbi.nlm.nih.gov/CBBresearch/Yu/midas/index.html (accessed on 16 February 2021)) was used to calculate monoisotopic masses. This report mainly used the NCBI online calculation under the following conditions: coarse-grained isotopic distribution mass accuracy (Da) = 1; fine-grained isotopic distribution mass accuracy (Da) = 0.01; and the polynomial-based method. Linear peptides directly adopted the amino acid sequences for core regions of *MSDIN* genes. Cyclized versions equaled the linear versions minus a water molecule. Besides cyclization, this study attempted to predict other possible posttranslational modifications, including hydroxylation, sulfoxidation, and bridge formation. One hydroxylation adds one O atom; known amatoxins can contain as many as four hydroxylations. Sulfoxidation adds an O atom, which only happens to the peptides containing Cys and Trp. The bridge formation between Trp and Cys loses two H atoms, resulting in a structure called tryptathionine [33]. How many and in which order these modifications would occur to a given *MSDIN*-related peptide are unknown. Our strategy was to try different combinations and look for matches in LC–HRMS.

### 2.7. LC–MS and LC–MS/MS Analyses

HPLC separation, as described above, was coupled with Agilent 6540 UHD precision mass quadrupole time-of-flight (Q-TOF) LC/MS (Agilent Technologies) equipped with an electrospray source. Eluant was monitored in positive electrospray ionization (ESI) mode with the capillary voltage at 3.5 kV. The drying gas (N_2_) temperature was 350 °C, and the flow rate was 8 L/min. The mass scan range was 500–1700 m/z. For collision energy in the subsequent MS/MS analysis, a range of 10–70 eV was applied.

After LC–MS/MS data were obtained, amino acid composition and combinations of candidate cyclic peptides were investigated first through the platform at the Center for Computational Mass Spectrometry (CCMS) based on the UniProtKB/Swiss-Prot database (https://proteomics.ucsd.edu/ProteoSAFe/ (accessed on 16 February 2021)). The workflow used MS-GF + (V1.3.0) on the website by extracting masses of fragments using Agilent Mass Hunter v8.0.0 (Agilent Technologies). The fragmentation diagrams were saved as Materials and Geometry Format (MGF) files and uploaded to the CCMS [34]. When no results were obtained with the CCMS, other platforms were included in the MS/MS analysis including Mascot (with NCBIprot database), pNove v3.1 [35], and XCMS online [36]. MS Convert (http://proteowizard.sourceforge.net/tools.shtml (accessed on 16 February 2021)) was used to obtain suitable file formats for each application.

All results from the above online tools were further confirmed or corrected via a thorough manual pipeline. First, high quality MS/MS spectra were obtained by optimizing the energy in the Q-TOF process. Best spectra were selected by choosing the ones with most y-type ions found in the fragmentations of their corresponding linear peptides. In some cases, increased amounts of samples were applied for optimal MS/MS spectra. Common fragmentation patterns for linear peptides were used because our preliminary results suggested that they were useful. Then, y-type ions for each MS/MS spectrum were determined based on core peptides of *MSDIN*s using Molecular Weight Calculator v6.50 (http://omics.pnl.gov (accessed on 16 February 2021)). Last, immonium ions were searched based on amino acid compositions. Only in situations in which most or all amino acids (by themselves or in the right combinations) in a given candidate cyclic peptide matched a *MSDIN* sequence did we deem the cyclic peptide real with the sequence designated by the core region of the *MSDIN* gene.

### 2.8. Cyclic Peptide Synthesis

All the predicted cyclic peptides found in this study were highly hydrophobic and difficult to chemically synthesize, but, fortunately, three of the cyclic peptides were successfully synthesized by Nanjing Peptide Biotech Ltd. (Nanjing, China). These synthesized cyclic peptides were dissolved in the same buffer as that used for extraction (absolute methanol:water:0.01 M hydrochloric acid = 5:4:1). The purity for each peptide was greater than 95% determined by HPLC and MALDI-TOF mass spectrometry. The MS/MS analysis described above was used to compare fragment ions from both natural and synthetic peptides.

## 3. Results

### 3.1. Amanita Genomes

Fruiting bodies of *Ae* and *Ar* were collected in excellent condition (Figure 1). For *Ae*, the N50 of the draft genome was 3.03 Mbp with a genome size of 49 Mbp (GC content = 46.6%), and the assembly contained 129 contigs. For *Ar*, the N50 was 690 kbp with the genome size at 45 Mbp (GC content = 45.99%), and the assembly comprised 134 contigs. The drafts for *As* and *Ap* were adopted from our previous report [18]. The Whole Genome Shotgun assemblies were deposited at DDBJ/ENA/GenBank as part of the accession JAEBUT000000000.

### 3.2. MSDIN Genes for Candidate Cyclic Peptides

With multiple known *MSDIN*s as queries, the genomes revealed the presence of *MSDIN* genes in the two newly sequenced *Amanita* species, i.e., *Ae* and *Ar*, yielding 23 and 39 predicted *MSDIN* genes, respectively (Table 1 and Table 2). With expanded query *MSDIN* sequences, four additional *MSDIN* genes were found in *As* compared to our previous report (Table 3), but none were found in *Ap*. In total, 66 *MSDIN*s were discovered, with 46 being novel. All genomes possessed *MSDIN* genes coding for at least three major cyclic peptide toxins, namely α-amanitin, β-amanitin, and phallacidin. Overall, the precursor genes share similar structures, designated as the leader peptide, core peptide, and recognition sequence [37]. Based on the newly discovered *MSDIN*s, linear and cyclic peptides with and without further modifications were predicted. A schematic diagram of the genome-guided approach is shown in Figure 2.

### 3.3. Transcriptome of As

*As* was chosen to be sequenced via the Illumina RNA-Seq technique, as it was a readily available species. In total, 11.7 Gb clean reads were obtained. The assembled transcriptome yielded 46.1 Mbp with 25,453 unigenes. A BLAST search against the transcriptome with *MSDIN*s from the *As* genome produced 22 *MSDIN*s (Table 3), four more than those in our previous result [18]. These four sequences were used as queries to re-search the genome, and the result confirmed their presence. Eighteen out of the 22 *MSDIN* genes were found to be expressed at the transcription level, i.e., 81.8% of the *MSDIN*s were expressed (Table 3). Regarding the gene structure, exons and introns were conserved among most of the *MSDIN* genes, while alternative splicing was detected in two of the *MSDIN* transcripts marked in grey in Table 3. The expressed *MSDIN*s were the focus for potential cyclic peptide production in the following MS and MS/MS analyses.

### 3.4. LC–HRMS and LC–MS/MS Analyses of Novel Cyclic Peptides in As

With the Agilent LC–HRMS platform, correlations between measured masses and predicted peptides based on genomic data were carefully assessed. No linear versions of these peptides were detected. Further, hydroxylation(s), sulfoxidation, and cross-bridging (by themselves or by various combinations) were not detected on predicted linear or cyclic peptides. However, two matches were found between theoretical and measured masses of two predicted cyclic peptides with no further modifications. These two matches corresponded to the cyclized core peptide sequences of TFLPPLFVPP (designated CylG1) and AFFPPFFIPP (named CylG2), respectively. The molecular formula for the candidate new cyclic peptide CylG1 is C_59_H_84_N_10_O_11_, and the theoretical molecular weight was found to be 1109.6394 [M+H]^+^. The measured molecular weight was 1109.6398 [M+H]^+^, with a mass discrepancy of 0.36 ppm (Figure 3A). The molecular formula for CylG2 is C_65_H_80_N_10_O_10_ with the theoretical molecular weight at 1161.6132 [M+H]^+^; the measured molecular weight was 1161.6161 [M+H]^+^ with a mass discrepancy of 2.50 ppm (Figure 3B). Two adduct ions of [M+Na]^+^ and [M+K]^+^ are also shown in the figure. CylG1 and CylG2 were treated as candidate new cyclic peptides for further characterization.

In order to determine the amino acid composition of CylG1 and CylG2, the candidate cyclic peptides were submitted to LC–MS/MS. The resulting spectra were first analyzed by the CCMS using the UniProtKB/Swiss-Prot database (Figure 3C). The calculated peptide sequence for CylG1 was PPLVFTPPLE. The amino acid composition differed at one position from the original sequence (E vs. F), which could be attributed to the application of a linear peptide database in the CCMS (Figure 3C underlined). The molecular weight of F was found to be 165.19, and the loss of water during cyclization resulted in a mass of 147.18, which was in agreement with the calculated mass of E (147.13) and therefore accounted for the presence of E over F. For Cy1G2, similar processes were applied, and a discrepancy was also found at the last predicted amino acid (underlined in Figure 3D). Some of our peptides did not return any results with the CCMS, and they were then analyzed with other platforms (Mascot, pNove, and XCMS). In general, the results were largely in line with the above. Though the automated pipelines offered some evidence, we mostly relied on the following manual process to determine the peptide sequences.

Manual amino acid composition analysis was mostly based on the y-type fragmentation of linear peptides. For CylG1, fragment ions were calculated using the core peptide, and the result of searching for these ions in the MS/MS spectrum is shown in Figure 3E. Every fragment ion was manually checked and confirmed. As illustrated in Figure 3E, y-type fragments (y2–y9) were found to be in strong agreement with the core peptide, and all the amino acids could be readily explained by the y-type ions. Immonium ions for P, L, and F were also identified for CylG1. As a result, CylG1 was assigned as a novel cyclic peptide with the amino acid composition and combinations matching the circular *MSDIN* core peptide sequence Cyclo (TFLPPLFVPP) (Figure 3C,E). CylG2 underwent the same analyses, in which case immonium ions for P, I, and F were identified. Similarly, y-type fragments (y2–y9) were shown to be in strong agreement with the core peptide, and all the amino acids complied with the y-type fragment ions (Figure 3D,F). In conclusion, CylG2 was confirmed to be a novel cyclic peptide with the sequence Cyclo (AFFPPFFIPP). The circular peptide sequence of CylG2 was found to be highly similar to antamanide, Cyclo (FFVPPAFFPP), differing by one amino acid (I vs. V).

### 3.5. Novel Cyclic Peptides in Other Amanita Species

CylG1 and CylG2 in *As* comprised our first effort towards finding novel cyclic peptides in the genus. Taking advantage of this genome-guided approach, we analyzed three other *Amanita* species, i.e., *Apa*, *Ae*, and *Ar*. In *Ar,* three new cyclic peptides with the sequences Cyclo (ISDPTAYP), Cyclo (FIPLGIITILP), and Cyclo (FPTRPVFP) were found (Appendix A), which were named CylH1, CylH2, and CylH3, respectively. In *Apa*, five new cyclic peptides with the sequences Cyclo (EFIVFGIFP), Cyclo (FVIIPPFIFP), Cyclo (YFFNDHPP), Cyclo (TIHLFSAP), and Cyclo (MHILAPPP) (Appendix A), which were named CylI1, CylI2, CylI3, CylI4, and CylI5, respectively. Further information on *MSDIN* genes in *Apa* are included in Appendix A. In *Ae*, two new cyclic peptides with the sequences Cyclo (FVFVASPP) and Cyclo (LFFPPDFRPP) were found (Appendix A), which were named CylJ1 and CylJ2, respectively. A previously known cyclic peptide called amanexitide with the sequence Cyclo (VFSLPVFF) [20] was also found in *A. exitialis* in this study (Appendix A). In conclusion, 12 novel cyclic peptides and one known cyclic peptide were discovered in the four sequenced species (Table 4). Relevant data analyses of mass spectrometry are included in Appendix A, Appendix A.

### 3.6. Confirmation of Three Cyclic Peptides via Synthetic Cyclic Peptides

Two of the novel cyclic peptides (CylG1 and CylG2) and amanexitide were chemically synthesized, and MS/MS data were obtained. The majority of fragment ions were shared between the natural and synthetic peptides in each pair (Figure 4). For CylG1, 16 primary fragment ions were consistent in both natural and synthetic compounds; for CylG2, 18 primary fragment ions were identical; and for amanexitide, 24 fragment ions were matched.

## 4. Discussion

Edible *Amanita* species are consumed in many parts of the world purposefully and by accident [6]. In the local markets of Yunnan Province, China, we constantly find at least three edible *Amanita* species. Unfortunately, many lethal *Amanita* species share morphological characteristics with the edible ones, leading to misidentification, consumption, and serious poisonings. In 2019, the Chinese Center for Disease Control and Prevention (China CDC) reported 276 independent mushroom poisonings in China with 769 patients and 22 deaths, and the majority of the deaths (20 cases) were caused by *Amanita* and amanitin-containing species [38]. The genome-guided approach adopted in this study identified 12 novel cyclic peptides and one known cyclic peptide in four *Amanita* species. The reliability of the results was furthered confirmed by comparing three of the natural peptides with synthetic ones. Our results indicated that there is a large cyclic peptide reservoir in *Amanita* species and provides a fast method of detection.

Traditional methods for identifying novel cyclic peptides in lethal *Amanita* species rely heavily on the availability of sufficient examined materials [1,6]. The MS-based method adopted in this study is much more sensitive, requiring only good MS/MS spectra. The first report using MS (as part of the approach) for identifying novel cyclic peptides in these mushrooms showed some initial promise [20], but the lack of genomic information limited its further application. An automated MS-based pipeline indicated that there are likely two novel cyclic peptides in *A. phalloides* [19], which pointed out a new direction, although manual analysis and further confirmation were lacking. In this study, manual analyses of MS and MS/MS data were shown to be the key for the pipeline. It is also worth noting that the MS analysis of the cyclic peptides required good sample preparation and separation. Obtaining optimal MS/MS spectra oftentimes needed tweaking with collision energy and sample amounts. Fragment ions could frequently only be recognized by manual inspection.

Most of the *MSDIN* genes were expressed at the transcription level in *As*, as suggested by the RNA-Seq data. Previous studies detected as many as half of *MSDIN*s expressed in *A. phalloides* and *A. bisporigera* [19], and additional expressed sequences were cloned from other *Amanita* species through RT-PCR [39,40]. These data indicated *MSDIN*s are mostly real, active genes that could potentially lead to production of many more cyclic peptides than previously known in the genus. Our results also strongly suggest that the amanitin biosynthetic pathway is highly versatile and capable of producing a wide range of cyclic peptides.

The cyclic peptides, including the toxins, are highly valuable resources. To date, most cyclic peptides in *Amanita* were first discovered from *A. phalloides* with a few exceptions [6]. It has been suspected that lethal *Amanita* species could produce significantly more cyclic peptides than what have been discovered [1,6]. The cyclic peptides discovered in this study lack further modifications besides cyclization and could therefore be readily synthesized for the purpose of screening for useful activities. Cyclic peptide pools in different species have distinctions, although the core set (major toxins) is relatively stable. As a result, the collection of the peptides in dozens of known lethal species in the genus should constitute a large cyclic peptide reservoir, potentially valuable for useful biological activities. α-Amanitin has been shown to have high potential in cancer research and treatment [41,42]. Phalloidin and phallacidin are valuable in cell research as generic molecular tools [43]. Antamanide functions as an antitoxin, reversing the effort of phalloidin [44], and researchers now know that antamanide inhibits the mitochondrial permeability transition pore by targeting the regulator cyclophilin D [24]. In addition, cycloamanides and related peptides possess potent activities useful in immunity-related medical situations [25,27]. Consequently, unknown cyclic peptides in *Amanita* species have the potential to bear useful activities.

Regarding the structures of the novel cyclic peptides, CylG1 and CylG2 have a characteristic amino acid composition with double Pro residues, no Trp and Cys, and without any further posttranslational modifications. These compounds and amanexitide share similarities to antamanide in that they contain ten amino acids (decapeptides) with a conserved sequence of XXXPPXXXPP; the two pairs of Pro residues are frequently found in *MSDIN*s. According to the MS/MS data, hydroxylation on the 12 evaluated peptides was not detected. Pro (100%), Phe (84.6%), and Ile (53.8%) were the most frequent residues in these cyclic peptides. CylG2 is very similar to antamanide, with only one amino acid difference (I vs. V). In *Ar*, the new peptide Cyclo (FIPLGIITILP), CylH2, has 11 amino acids, which is by far the longest in all discovered *Amanita* cyclic peptides. Despite some structural similarities, the biological activities of these peptides require further study.

## 5. Conclusions

In this report, two deadly *Amanita* species (*A. rimosa* and *A. exitialis*) were sequenced, and, in total, 46 novel *MSDIN* genes discovered in four draft genomes. A majority of the *MSDIN* genes were found to be expressed at the mRNA level in *A. subjunquillea*. Using mass spectrometry, 12 novel cyclic peptides, which are potentially valuable for useful activities, were identified. The results were further confirmed by comparing three natural peptides with synthetic ones. In conclusion, the results suggest that the genome-guided approach is robust for identifying novel cyclic peptides in lethal *Amanita* species.

## Figures and Tables

**Figure 1 jof-07-00204-f001:**
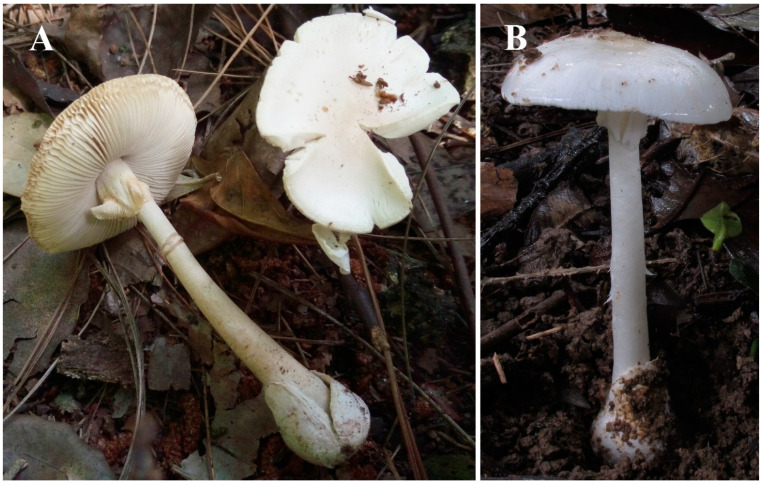
Deadly *Amanita* species for genome sequencing and peptide analyses. (**A**) *Amanita rimosa*. (**B**) *Amanita exitialis*.

**Figure 2 jof-07-00204-f002:**
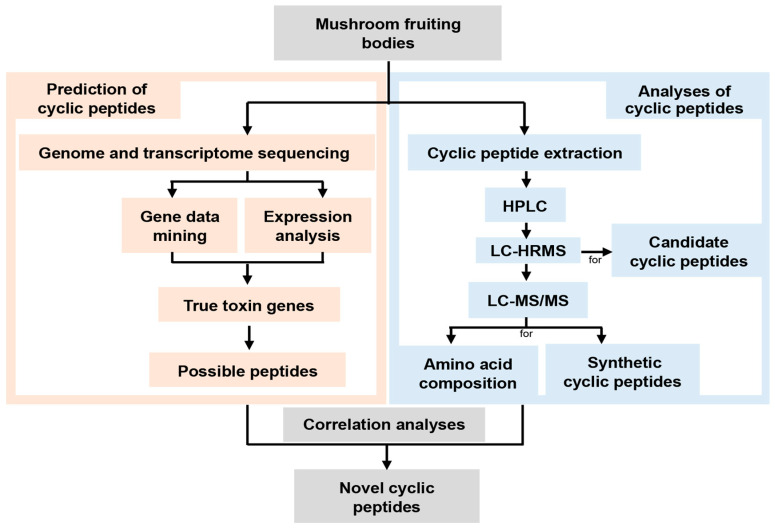
A flowchart for characterizing cyclic peptides in *Amanita* species. HRMS: high resolution MS.

**Figure 3 jof-07-00204-f003:**
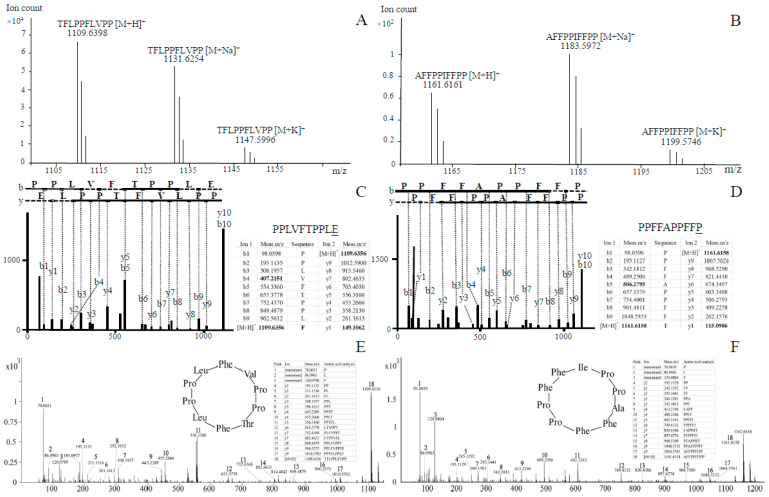
LC–HRMS and the amino acid composition of the cyclized core peptide sequences of TFLPPLFVPP and AFFPPFFIPP (CylG1 and CylG2, respectively) in *Amanita subjunquillea*. Positive LC–HRMS spectra of CylG1 (**A**) and CylG2 (**B**), with [M+Na]^+^ and [M+K]^+^ adduct ions indicated. (**C**,**D**) Amino acid composition analysis through the Center for Computational Mass Spectrometry (CCMS) for CylG1 and CylG2, with b- and y-type fragment ions shown (bold fonts were manually added for ions not recognized by the website); predicted linear peptide sequences are shown above the inserted tables (the last amino acids are underlined as they were erroneous). (**E**,**F**) Manual analyses of recognized fragment ions for CylG1 and CylG2 with inserted tables of matched amino acids; immonium ions are shown when present. The determined cyclic peptides are shown as circles.

**Figure 4 jof-07-00204-f004:**
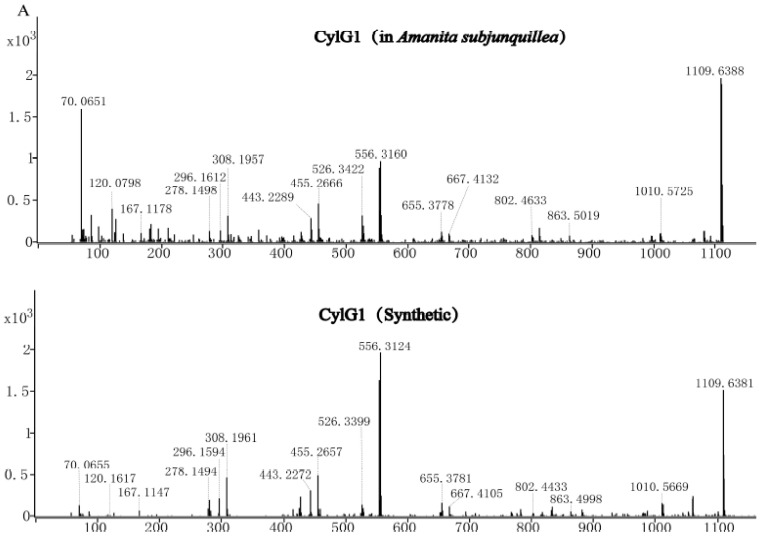
MS/MS fragment comparison of three natural and synthetic cyclic peptides from *Amanita* species. Fragment ions of natural and synthetic CylG1 (**A**), CylG2 (**B**), and amanexitide (**C**). CylG1 and CylG2 are produced by *A. subjunquillea*, and amanexitide is produced by *A. exitialis*. Shared ions are labeled.

**Table 1 jof-07-00204-t001:** The MSDIN gene family in *Amanita rimosa.*

Leader Peptide	Core Peptide	Recognition Sequence	Monoisotopic Mass
MSDINSTRLP	**IWGIGCNP**	SVGDEVTALLTRGEA	918.3541 (α-amanitin)
MSDINATRLP	**IWGIGCNP**	SVGDEVTALLASGEA	918.3541 (α-amanitin)
MSDINATRLP	**IWGIGCDP**	CVGDDVAALTTRGEA	919.3382 (β-amanitin)
MSDINATRVP	**AWLVDCP**	CVGDDISRLLTRGEK	846.3217 (phallacidin)
MSDINATRLP	**AWDSKHP**	CVGDDVSRLLTRGE	821.3820
MSDINATRLP	**AWDSKHP**	CVGDDISRLLTRGE	821.3820
MSDINATRVP	**AWLAECP**	CVGDDISHLLTRGE	770.3421
MSDINASRLP	**FFIIIVKP**	CGNPYVSDDVNSTLTRGE	957.6052
MSDINTSRLP	**FIPLGIITILP ** **^★^**	CVSDDVNTTITRGD	1177.7475
MSDINTACLP	**FLFPVIPP**	CLSEDANVVVLNSGE	910.5317
MSDINVTRLP	**FFPIVFIPP**	CI	1057.6000
MSDINIARLP	**IFWFIYFP**	CVGDDVDNTLSRGE	1113.5688
MSDINVTRLP	**IFLIMFIPP**	CIGDDAASILKQGE	1071.6191
MSDINTSCLP	**IFIAFPIPP**	CVSDDIQTVLTRGE	995.5844
MSDTNTACLP	**IFIAFPIPP**	CVSDDIQTVLTRGE	995.5844
MSDINASRLP	**ILKKPWAP**	SVCDDVNSTLTRGE	933.5800
MSDINVARLP	**ISDPTAYP ** **^★^**	CVGDDIQAVVKRGE	844.3967
MSDINATRLP	**IIIVLGLIIP**	LCVSDIEMILTRGE	1044.7311
MSDINASRLP	**IILAPIIP**	CISDDVNTTLTCAE	830.5630
MSDINTTGLP	**HFYNLMPP**	CFSDDTGMVLVRGE	999.4637
MSDINATRLP	**HPFPLGLQP**	CAGDVDNFTLIKGE	986.5338
MSDINASCLP	**LILVANGMAYV**	--SDDVSPTLTRGE	1144.6315
MSDINTARLP	**SYIPFPPP**	CLSEDTNAVLMLGE	898.4589
MSDINTARLP	**SYIPFPPP**	CLSEDTNAVLMLGE	898.4589
MSDINTSRFP	**SYGYRAFP**	CVGDDVEMVLMHGE	941.4396
MSDINVTRLP	**VLVFIFFLP**	CISDDAASIIKLGE	1075.6470
MSDIDTTRLP	**LILFTLQP**	SIGDDVNPTLTRGEK	925.5637
MSDIHAARLP	**FPTRPVFP ** **^★^**	SAGDDMIEVVLGRGE	941.5123
MSDNNAARLP	**FYFYLGIP**	SDDAHPILTRGERLA	1000.5058
MSDTNTARLP	**ILFIQLEIP**	CISDDVHPVLTRGE	1066.6427
MSDVNTTRLP	**FNFFRFPYP**	CICDDSEKVLELGE	1215.5866
MSEINTARFP	**NHGHRTIP**	CVGDDIEMVLMHGE	912.4678
MSEINTSRLP	**LVFIPPYFAP**	CVSDDIQMVLTLGE	1144.6321
MFDMNTTCLP	**GFIIYAYV**	--GDDVNHTLTRGE	926.4902
MLDINTARLP	**FSLPTFPP**	CVSDEIDVVLKRGE	886.4589
MLDINATRFP	**LGRPTHLP**	CVGDDVNYIL	871.5028
MTDINDARLP	**ILLLIFFWIP**	CANDDDENILNRG	1255.7733
MTDINDTRLP	**FVWILWLWLA**	CVGDDTSILNRGE	1327.7481
MPDINVTRLP	**LLIIVLLTP**	CISDDNNILNRGK	975.6732

^★^ and underlined letters indicate novel cyclic peptides detected with MS and MS/MS. The monoisotopic masses are for unmodified cyclic peptides based on MSDIN core peptides, except for major toxins (α-amanitin, β-amanitin and phallacidin). Core peptides are indicated in bold.

**Table 2 jof-07-00204-t002:** The MSDIN gene family in *Amanita exitialis.*

Leader Peptide	Core Peptide	Recognition Sequence	Monoisotopic Mass
MSDINATRLP	**IWGIGCNP**	CVGDDVTSVLTRGEA	918.3541(α-amanitin)
MSDINATRLP	**IWGIGCDP**	CVGDDVTALLTRGEA	919.3382(β-amanitin)
MSDINATRLP	**AWLVDCP**	CVGDDVNRLLTRGE	846.3217(phallacidin)
MSDINATRLP	**AWLTDCP**	CVGDDVNRLLTRGE	786.3371
MSDINTTRLP	**FVFVASPP ** **^★^**	CVGDDIAMVLTRGE	844.4483
MSDINTARLP	**FIWVFGIP**	--GDDIGTVLTRGEK	959.5269
MSDINLTRLP	**GIIAIIP**	CVGDDDDVNSTLTRGQ	677.4476
MSDINATRLP	**IILAPVIP**	CISDDNDP--TLTRGQ	816.5473
MSDINTARLP	**IPIPPFFFP**	FVSDDIEIVLRRGEK	1055.5844
MSDINTARLP	**IPIPPFFFP**	FVSDDIEIVLRRGEK	1055.5844
MSDINATRLP	**IGRPQLLP**	CVGGDVNYILISGEK	874.5389
MSDINPTRLP	**IFWFIYFP**	CVSDVDST-LTRGE	1113.5688
MSDINTARLP	**IYRPPFYALP**	CVGDDIQAVLTRGE	1217.6597
MSDINTARLP	**IIWIIGNP**	CVSDDVERILTRGE	906.5327
MSDINVIRAP	**LLILSILP**	CVGDDIEV-LRRGE	862.5892
MSDINATRLP	**LFFPPDFRPP ** **^★^**	CVGDADNFTLTRGEK	1213.6284
MSDINATRLP	**LFFPPDFRPP ** **^★^**	CVGDADNFTLTRGE	1213.6284
MSDINVIRLP	**SMLTILPP**	CVSDDASNTLTRGE	852.4779
MSDINTARLP	**VFSLPVFFP ** **^★^**	--SDDIQAVLTRGE	1033.5637
MSDINVTRLP	**VFIFFFIPP**	CVGDGTADIVRKGEK	1107.6157
MSDINATRLP	**VWIGYSP**	CVGDDCIALLTRGE	802.4014
MSDINATRLP	**VWIGYSP**	CVGDDCIALLTRGE	802.4014
MTDINDTRLP	**FIWLLWIWLP**	SVGDD-NNILNRGEE	1367.7794

^★^ and underlined letters show novel and known cyclic peptides detected with MS and MS/MS. The monoisotopic masses are for unmodified cyclic peptides based on MSDIN core peptides, except for major toxins (α-amanitin, β-amanitin and phallacidin). Core peptides are indicated in bold.

**Table 3 jof-07-00204-t003:** Expression of the MSDIN gene family from *Amanita subjunquillea.*

Leader Peptide	Core Peptide	Recognition Sequence	Monoisotopic Mass	Expressed
MSDINATCLP	**IWGIGCNP**	CVGDEVAALLTRGEALC	918.3541(α-amanitin)	√
MSDINATRLP	**IWGIGCDP**	CVGDEVTALLTRGEALC	919.3382 (β-amanitin)	√
MSDINATRLP	**IWGIGCDP**	CIGDDVTALLTRGEALC	919.3382 (β-amanitin)	√
MSDINATRLP	**AWLATCP**	CAGDDVNPTLTRGESLC	788.3160 (phalloidin)	√
MSDINATRLP	**AWLVDCP**	CVGDDINRRVVSAFA-C	846.3217 (phallacidin)	√
MSDMNATRLP	**LIQRPFAP**	CVSDDVDFALIRRCALVYAESSV	922.5389	√
MSDINTARLP	**HFASFIPP**	CIGDDIEMVLKRGESLC	896.4545	√
MSDINTARLP	**TFLPPLFVPP ** **^★^**	CVSDDIEMVLTRGESLC	1108.6321	√
MSDINATRLP	**LNILPFMLPP**	CVGDDVNPTLTRGEDLC	1135.6464	√
MSDMNATRLP	**LIQRPYAP**	CVSDDVNSPLTRGESLC	938.5338	√
MSDINTARLP	**IGRPESIP**	CVGDDIEMILERGQKLC	849.4709	√
MSDINTARLP	**LRLPPFMIPP**	CVGDDIGMVLTRGENLC	1161.6733	√
MSDVNATRLP	**FNFFRFPYP**	CIGDDSASVLGLGESLC	1215.5866	√
MSDINATRLP	**SSVLPRP**	CVGDVDNIILTSREKLC	736.4232	√
MSDINTARLP	**AFFPPFFIPP ** **^★^**	CVSDDIEMVLTRGESLC	1160.6059	√
MSDINATRLP	**IPILPIPP**	YCSDDANTTLTLGESLC	840.5473	√
MSDINATRLP	**LFLLAALGIP**	--SDDADSTLTRGESLC	1008.6372	√
MSDTNDARLP	**LFFWFWFLWP**	SVSDDIDSVLNRGEDLC	1469.7325	√
MSDMNVARLP	**ISDPTAYP**	CVGGDIHAVLRRGE	844.3966	×
MSDMNVARLP	**ISDPTAYP**	CVGGDIHAVLRRGE	844.3966	×
MSDINVTCLP	**FIFWFFWPP**	CVGDDAASIIK-GK	1267.6218	×
MSDINAARLP	**FIFPPFFIPP**	CVSDDIEMVLTRGE	1202.6528	×
MSDINTVCLP	**LQKPWSRP**	CVGDDIEMILERGE	992.5556	×
MFDINITRLP	**IFWFIYFP**	CVGDDVTALLTRGE	1113.5689	×

^★^ and underlined letters indicate novel cyclic peptides detected with MS and MS/MS. The monoisotopic masses are for unmodified cyclic peptides based on MSDIN core peptides, except for major toxins (α-amanitin, β-amanitin, phalloidin and phallacidin). Grey letters indicate alternative splicing. Core peptides are indicated in bold.

**Table 4 jof-07-00204-t004:** Twelve novel cyclic peptides and one known cyclic peptide discovered in *Amanita subjunquillea*, *A. rimosa*, *A. exitialis*, and *Amanita pallidorosea.*

Species	Cyclopeptide Sequence	Molecular Formula	Theoretical (m/z)	Measured (m/z)	δ (ppm)
*A. subjunquillea*	TFLPPLFVPP (CylG1)	C_59_H_84_N_10_O_11_	1109.6394	1109.6398	0.36
	AFFPPFFIPP (CylG2)	C_65_H_80_N_10_O_10_	1161.6132	1161.6161	2.50
*A. rimosa*	ISDPTAYP * (CylH1)	C_39_H_56_N_8_O_13_	845.4039	845.4040	0.12
	FIPLGIITILP (CylH2)	C_61_H_99_N_11_O_12_	1178.7547	1178.7555	0.68
	FPTRPVFP (CylH3)	C_48_H_67_N_11_O_9_	942.5196	942.5191	0.53
*A. pallidorosea*	EFIVFGIFP (CylI1)	C_56_H_75_N_9_O_11_	1050.5658	1050.5694	3.43
	FVIIPPFIFP (CylI2)	C_65_H_90_N_10_O_10_	1171.6914	1171.6941	2.31
	YFFNDHPP (CylI3)	C_51_H_59_N_11_O_12_	1018.4417	1018.4421	0.39
	TIHLFSAP (CylI4)	C_42_H_62_N_10_O_10_	867.4723	867.4733	1.15
	MHILAPPP (CylI5)	C_41_H_64_N_10_O_8_S	857.4702	857.4714	1.40
*A. exitialis*	FVFVASPP (CylJ1)	C_44_H_60_N_8_O_9_	845.4556	845.4582	3.08
	LFFPPDFRPP ^#^ (CylJ2)	C_63_H_83_N_13_O_12_	1214.6357	1214.6357	0.00
	VFSLPVFFP ^^^	C_56_H_75_N_9_O_10_	1034.5709	1034.5734	2.42

* indicates that the sequence has been found in *A. subjunquillea*, *A. pallidorosea,* and *A. rimosa*. ^#^ indicates this sequence has two copies. ^^^ indicates amanexitide.

## Data Availability

The Whole Genome Shotgun assemblies were deposited at DDBJ/ENA/GenBank as part of the accession JAE-BUT000000000.

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
