# Peer review of "Novel Cyclic Peptides from Lethal Amanita Mushrooms through a Genome-Guided Approach"

_jof, 2021, doi:10.3390/jof7030204_

Round 1

Reviewer 1 Report

The publication entitled "Novel cyclic peptides from lethal Amanita mushrooms through a genome-guided approach" is a well-prepared, clarified, research work aimed at using mass spectrometry techniques and genome analysis to detect new cyclic peptides in representatives of the  Amanita genus. This is particularly important due to the large biomedical potential of such peptides and the representatives of the kingdom of fungi as such. The manuscript is well organized and the experiments are carefully planned. It was hard for me to find any major clear substantive errors.

However, I have a few comments:
The method of extracting cyclic peptides requires a literature source i.e. Enjalbert, F. , Gallion, C. , Jehl, F. , Monteil, H. , & Faulstich, H. (1992). Simultaneous assay for amatoxins and phallotoxins in Amanita phalloides Fr. by high‐performance liquid chromatographyJournal of Chromatography598, 227–236 and Hallen, H. E. , Watling, R. , & Adams, G. C. (2003). Taxonomy and toxicity of Conocybe lactea and related speciesMycological Research107, 969–979.  Or similar, of course, taking into account the fact that the procedures have been significantly modified by the authors.

Best regards

Author Response

Response to reviewer 1,

We sincerely thank reviewer 1 for the positive review! Regarding the one suggestion, we did use the same extraction buffer and the two refs are now included in the Materials and Methods section.

Sincerely,

Hong Luo

Key Laboratory for Plant Diversity and Biogeography of East Asia

Kunming Institute of Botany

Chinese Academy of Sciences

Kunming 650201, Yunnan, China

E-mail: luohong@mail.kib.ac.cn

Work Phone (and Fax): 86-871-65223508

Reviewer 2 Report

The article is potentially written deciphering the “Novel cyclic peptides from lethal Amanita mushrooms through a genome-guided approach” by Zhou et al. However, the manuscript has to be improved to make it more suitable for acceptance in the journal. The following questions needs to be addressed.

Comment 1: Please include line numbers, it will be easy to review

Comment 2: In abstract, when you write first time, please include both genus and species name, for at least one species.

Comment 3: What is “MSDIN”, give abbreviation.

Comment 4: Write abbreviation MS-MS in abstract.

Comment 5: Rewrite conclusion both in Abstract and Conclusion.

Comment 6: In Introduction, paragraph 3, please include abbreviations for TLC, HPLC and NMR

Comment 7: Lot of information in Results section, did not find detailed discussion. Please improve discussion with previous research reports.

Comment 8: Please check English throughout the manuscript.

Author Response

Response to reviewer 2,

We are grateful for reviewer 2 for the suggestions! All comments were dealt with carefully, please see the responses below.

Comment 1: The missing line numbers were due to Word file conversion and have been addressed by the journal’s office.

Comment 2: Full name for the species is provided in the abstract.

Comment 3: “MSDIN” is described as “named after the first five amino acid residues of the precursor peptides”.

Comment 4: Full name and abbreviation of MS/MS are provided in the abstract.

Comment 5: Relevant parts have been rewritten in both Abstract and Conclusion.

Commnet 6: Full names followed by abbreviations for TLC, HPLC and NMR are provided.

Comment 7: A new paragraph has been added to cover main information of the Results section. Previous researches regarding MS-based approach are included.

Comment 8: A native speaker was involved in the process. Many minor changes throughout the manuscript were made, please see the line numbers below.

Line numbers for minor changes: 58, 63, 68, 70, 72-74, 110, 121, 134, 137, 140, 146, 152-153, 169, 174, 184-187, 204, 230, 244, 252, 259-260, 275, 280, 336-337, 355-356, 361-380, 391-393, 397-406, 416-417, 440, 458, 466, 468, 472-474, 476-478, 480-481, 483, 489, 491-492, 495, 498-499, 501, 503, 505, 507-508, 511-516, 518-519, 521, 523, 525-531, 534, 536, 538-539, 541, 543, 546-548.

Sincerely,

Hong Luo

Key Laboratory for Plant Diversity and Biogeography of East Asia

Kunming Institute of Botany

Chinese Academy of Sciences

Kunming 650201, Yunnan, China

E-mail: luohong@mail.kib.ac.cn

Work Phone (and Fax): 86-871-65223508